# The Identification of Novel Biomarkers Is Required to Improve Adult SMA Patient Stratification, Diagnosis and Treatment

**DOI:** 10.3390/jpm10030075

**Published:** 2020-07-29

**Authors:** Piera Smeriglio, Paul Langard, Giorgia Querin, Maria Grazia Biferi

**Affiliations:** 1Centre of Research in Myology, Institute of Myology, Sorbonne Université, INSERM, 75013 Paris, France; p.langard@institut-myologie.org (P.L.); g.querin@institut-myologie.org (G.Q.); 2Association Institut de Myologie, Plateforme Essais Cliniques Adultes, 75013 Paris, France; 3APHP, Service de Neuromyologie, Hôpital Pitié-Salpêtrière, 75013 Paris, France

**Keywords:** spinal muscular atrophy, adult patients, disease heterogeneity, Nusinersen, disease modifiers, functional outcomes, biomarkers, epigenetic changes, -omics approaches

## Abstract

Spinal muscular atrophy (SMA) is currently classified into five different subtypes, from the most severe (type 0) to the mildest (type 4) depending on age at onset, best motor function achieved, and copy number of the *SMN2* gene. The two recent approved treatments for SMA patients revolutionized their life quality and perspectives. However, upon treatment with Nusinersen, the most widely administered therapy up to date, a high degree of variability in therapeutic response was observed in adult SMA patients. These data, together with the lack of natural history information and the wide spectrum of disease phenotypes, suggest that further efforts are needed to develop precision medicine approaches for all SMA patients. Here, we compile the current methods for functional evaluation of adult SMA patients treated with Nusinersen. We also present an overview of the known molecular changes underpinning disease heterogeneity. We finally highlight the need for novel techniques, i.e., -omics approaches, to capture phenotypic differences and to understand the biological signature in order to revise the disease classification and device personalized treatments.

## 1. Introduction

Spinal muscular atrophy (SMA) is a neurodegenerative disease affecting motoneurons (MN) in the brainstem and spinal cord caused by the homozygous mutation of the Survival of Motor Neuron 1 (*SMN1*) gene [1]. The disease presents with a wide spectrum of clinical severity and patients are classified into five types, depending on the age of onset and motor milestones achieved. Patients with type 0 SMA are the most extreme cases with death early after birth. Severity decreases from type 1 to type 4 SMA, which only presents mild symptoms starting at the adult age and have a very slow evolution over time (reviewed in [2], see Table 1 and Figure 1A). Attempts to explain this phenotypic variability have been made and several disease modifiers are known [3]. The most widely accepted modifier is the copy number of the *SMN2* gene, a centromeric paralog of *SMN1*. Following a cytosine to thymine transition in exon 7, which creates an exon splicing suppressor (ESS), the transcription of *SMN2* leads to 90% of transcripts coding for a truncated SMN protein. The remaining 10% produces a full-length (FL) SMN protein, thus inducing low levels of SMN expression.

*SMN2* copy number is directly correlated with the SMN expression level; therefore, a higher copy number is associated with a milder phenotype [4]. *SMN2* copy number is used, together with age of onset and motor abilities achieved, as an additional parameter to stratify SMA patients (Table 1 and Figure 1B). This additional factor is, however, still insufficient to explain the phenotypic variability among siblings carrying the same mutation in *SMN1* and same *SMN2* copy number [5,6].

In the last few years, other gene modifiers have been described, such as Plastin 3 (PLS3) [8], Coronin 1C (*CORO1C*) [9], and Neurocalcin Delta (*NCALD*) [10] (Figure 1C). It remains puzzling to note that the expression changes of these genes do not fully explain the phenotypic heterogeneity in the SMA population. Additional modifier conditions to be considered are (i) the environmental factors that may influence the final phenotype of the patients [11] and (ii) the differential vulnerability of MN subtypes that is affected by the level of SMN protein [12,13]. Although all of these factors are known contributors to the differential disease severity, other unknown aspects might need to be considered in the future for a comprehensive description of the disease.

Recent advances in the SMA field have led to the development of therapeutic approaches, aiming to increase the level of SMN protein targeting *SMN2* through an antisense (AS) oligonucleotide [14,15] or a pre-mRNA splicing modifier drug [16], as well as inducing the restoration of the SMN protein via gene therapy [17].

Approved by the Food and Drug Administration in 2016, Nusinersen is an AS oligonucleotide that targets exon 7 of *SMN2* to facilitate its inclusion and trigger a higher production of the full-length SMN transcript. Nusinersen is administered via intrathecal injection, directly into the cerebrospinal fluid (CSF), in a two-phase treatment: (1) a loading phase of four doses 14, 14, and 30 days apart followed by (2) a maintenance phase with a dose every four months. Treatment with Nusinersen showed very promising results in young type 1 patients [14,15] and some positive effects in type 2 and 3 patients (no type 4 patients have been enrolled in clinical trials to date) [18,19,20]. Unexpectedly, the latter studies which include adult SMA patients (>18 years old) showed that their response to Nusinersen, evaluated as motor function improvement, was highly variable, with 40–50% of responders at best [18,19,20]. These observations suggest that factors modulating response to treatment, in particular for the older patients, remains to be uncovered. Moreover, due to the lack of natural history data and the higher heterogeneity among the adult SMA patient population, therapeutic monitoring of these patients is particularly challenging [21,22]. Therefore, an exceptional effort is needed to facilitate further characterization of the clinical and molecular profiles of these SMA patient populations.

In this context, the identification of novel biomarkers and precise methods for functional outcomes evaluation are unmet needs. Reaching these goals will provide the means to (i) redefine patients’ classification, (ii) characterize the full molecular profile of therapy responders or non-responders, and (iii) determine inclusion criteria in clinical trials and treatment protocols. Therefore, several types of biomarkers have been evaluated in order to capture patients’ pre-existing differences and their response to treatment.

As previously reviewed, biomarkers can be classified as biomolecular, mirroring the molecular changes, and functional to picture the phenotypic disease progression [23]. This review will initially focus on the most promising biomarkers for adult SMA (>18 years old). Data on therapeutic monitoring are gathered through the different studies reporting results from treatment with Nusinersen, the most widely used therapy for SMA worldwide up to date. Current available information on the molecular signature of SMA patients will be then presented, including gene modifiers, epigenetic changes, and putative molecular biomarkers. Furthermore, although various molecular biomarkers have been proposed in the last few years as possible candidates, no factor has been proven to date to faithfully follow the disease progression in all SMA types. Therefore, here we will also discuss the potential of unbiased -omics (e.g., proteomics) approaches as valuable tools for the identification of novel SMA-specific biomarkers.

In perspective, an in-depth analysis of both functional outcomes’ evaluation and molecular biomarkers, would help determine the differences that define the wide spectrum of clinical features among SMA patients and their disparate response to treatment. Efforts in this direction could lead to the development of personalized medicine approaches in the future.

## 2. Evaluation of Functional Outcomes in Adult Patients

Functional outcomes are essential to monitor disease progression and to precisely understand where the patient stands in the wide spectrum of the disease phenotypes. For chronic diseases, the longitudinal study of functional outcomes contributes to define the natural history of the disease and can also be used to monitor the therapeutic response. In SMA, several tests have been used for diagnosis, prognosis, or therapeutic monitoring. These tests have frequently turned out to be unadaptable to the adult SMA patient population (>18 years of age) due to its high heterogeneity. This group of patients still needs a precise characterization and patient-specific therapeutic adjustments or development. Therefore, this section of the review will focus on functional outcomes reported to reliably assess the Nusinersen response in SMA adult patients. Data collected in younger patients will be presented as comparison.

Gradual motor function loss is a common feature of all SMA types. While adult patients are generally classified as less severe cases than type 1 children, their motor function also declines progressively [24]. The motor abilities can be monitored by a variety of measurements, spanning from scales assessing the general mobility or muscle function, to the more global walking analysis. Respiratory function is also frequently assessed in adult SMA patients, as respiratory failure is still the most frequent cause of morbidity in SMA patients [25]. In this section, we are documenting functional outcomes to assess (i) general mobility; (ii) motricity of the upper or lower limbs, in conjunction with muscle strength or fatigue; (iii) level of ambulation; and (iv) respiratory function in adult SMA patients.

### 2.1. General Mobility Tests

The Hammersmith Functional Motor Scale Expanded (HFMSE) is a functional scale used for the assessment of physical abilities. It was adapted from the classic HFMS adding 13 new items that allow to capture a wider range of motor skills [26]. Therefore, the HFMSE has been proven to be a reliable indicator of the wide arrays of movements in adult SMA patients types 2 and 3. In the effort to correlate the HFMSE score with the adult SMA types and subtypes, we collected the most recent available data on SMA types 2 and 3 (Table 2).

Independent studies exploring this aspect showed a consistent range of HFMSE score: between 0 and 20 for type 2 and between 25 and 66 for type 3 patients [27,28,29]. A larger range was recorded for the type 3 patients followed up by Walter and colleagues [18]. A similar trend, but with lower HFMSE score, for type 3b patients (with symptoms onset after 3 years of age) has been described very recently by Kessler et al. [30]. Despite the variability for subtypes 3 and 3b, HFMSE seems to be a reliable indicator of the differences among SMA types.

Furthermore, this scoring system showed that there was a progressive gain during the course of the Nusinersen treatment, with an average gain of 3.12 points at 14 months after the first injection in 57 patients [19]. Results differed between SMA types 2 and 3, with a higher average HFMSE increase for SMA type 3 patients. Interestingly, no correlation was observed between the age at treatment and the evolution of the functional score during the treatment [19,29].

### 2.2. Revised Upper Limb Module

The revised upper limb module (RULM) (see Table 3), a reconsidered version of the original upper limb module (ULM) [32], has been created to measure upper limb function in a wider range of patients [33] than the non-ambulatory young children and weaker patients [32]. The RULM includes 20 tasks based on a scale of 3 scores: 0 (unable), 1 (able, with modification), and 2 (able, no difficulty) [33]. A large natural history study across three countries analyzed the RULM score for 114 SMA type 2 and 3 patients, ranging from weak non-ambulant to stronger ambulant, over the course of 12 months [34]. This analysis confirms that the RULM scale can detect a wide spectrum of upper motor abilities even in ambulant patients where the ceiling effect was rarely reached (11.4% of the cohort), overcoming the ULM scale limitations. Furthermore, it has been demonstrated that the RULM score is more useful than the HFMSE to picture motor differences in wheelchair-dependent adult SMA type 2 and 3 patients [35], suggesting that it can be used as diagnostic marker. Moreover, in patients treated with Nusinersen, RULM was shown to be more accurate than the HMFSE in detecting the improved motor abilities of type 2 and 3 patients (both young and adults) treated with Nusinersen [18,19] at later time points (10–14 months after the first injection). All these data demonstrate that the RULM scale is a sensitive marker for both diagnostic and therapeutic monitoring purposes and it could be used in combination with a more general scale to reveal the finest motor differences for the large spectrum of adult patients.

### 2.3. Quantitative Assessment of Ambulation Capacity

Among the five SMA types, only type 3 and 4 patients can walk unassisted. This is the highest milestone achievable by these patients; however, the probability of remaining ambulant decreases with aging. According to a study of natural history for type 3 SMA patients published by Zerres et al. [37], the probability of preserving ambulation throughout life is tightly linked to the age of disease onset. The authors showed that 83.7% of patients (n = 72) with symptomatic appearance between 18 and 35 months of age—corresponding to SMA type 3a—are able to walk 10 years after disease onset, and this percentage plummeted to 30% at 25 years after disease onset. Accordingly, among patients with a disease onset between 3 and 15 years old (n = 109)—corresponding to SMA type 3b subset—an average 95% were able to walk 10 years after disease onset and 75% at 25 years. These data clearly showed that the loss of the ability to walk declines slowly in patients with a later disease onset, suggesting that this outcome can be monitored to assess disease progression. Several functional scales have been exploited to strictly monitor the ability to walk, such as the 6-min walking test (6MWT).

The 6MWT is a functional test designed for ambulant patients, aiming to measure the distance that a patient can cover during a six-minute timelapse. The test can also document the gait pace by minute and indirectly describes motor fatigue over time. Guidelines for this test were published in 2002 by the American Thoracic Society (ATS) [38]. The 6MWT has been approved as an endpoint outcome in SMA patients motor function assessment [39]. As described in the ATS statement for the 6MWT, individual factors might influence the outcome of the test, including sex, age, height, and weight, but also the personal motivation of the patient. For healthy subjects between 10 and 59 years of age, the 6MWT values range approximately from 600 to 850 meters (m) [40]. In SMA patients, the distance covered in 6 minutes is affected by both age and the type of SMA. Montes et al. [28] showed that the median value is generally higher for SMA subtype 3b averaging at 368 m (n = 28) compared to 253 m for subtype 3a (see Table 4). However, the mean rate of progression in the two SMA subtypes did not differ significantly after 1 years from the baseline obervation. Moreover, one can notice the internal variability into each subtype and the important overlap between subtypes 3a and 3b.

The analysis of the ambulation is a valuable tool to follow up the response to treatment and few studies reported data obtained with the 6MWT. Recently, Hagenacker et al. [19] published the results of a multicenter, observational study on a cohort of 173 adult SMA patients treated with Nusinersen and followed up to 14 months. The 6MWT was used as a secondary endpoint for ambulant patients (n = 46 at baseline; see Table 4). At all time points analyzed (6, 10, and 14 months) patients showed a significant increase in the outcome of the 6MWT with an average gain of 22.1 m (8.7 m–35.6 m; 95% Confidence Interval) after 6 months and 46 m (25.4 m–66.6 m; 95% C.I.) after 14 months. Very interestingly, along the course of treatment, few non-ambulatory patients gained the ability to walk. This large cohort study provides evidence for the efficacy of Nusinersen treatment to improve the ambulatory function in adult SMA patients.

While the 6MWT gave some indications about treatment response in walking adult SMA patients, the variability in the degree of ambulation at both baseline and after treatment should be further explored and correlated to other functional and molecular parameters to better understand the disease.

### 2.4. Lung Function Tests

Respiratory function is impaired in SMA patients as the diaphragm and intercostal muscles are affected by the progression of the disease. The association of this alteration with recurrent scoliosis in patients can result in restrictive lung disease. Difficulty to cough can contribute to a reduced clearance and facilitate the onset of respiratory infections. This decline in the respiratory function is associated with mortality and morbidity in SMA patients of all types [14,25]. For this reason, many efforts have been devoted in the last few years to the search for appropriate parameters to assess this function.

The natural history study of lung function in SMA patients, reported by Wijngaarde et al. [25], analyzed the Forced Expiratory Volume in 1 second (FEV${}_{1}$) in a cohort of 170 patients SMA types 1c to 4. Using these data, the authors were able to build age-dependent models of the evolution of this measure for 131 patients—subtypes 1c to 3b. The linear models obtained, as expected, showed a progressive annual decrease of the FEV${}_{1}$ from 100.35% predicted in newborns subtype 3b—almost normal lung function—to 42.12% in 1c newborns. They reported a 1.29% and 1.37% annual rate of decline for subtypes 2a and 2b, respectively, which was associated with an early start of mechanical ventilation (median 12.3 and 16.8 years old, respectively). Subtypes 3a showed a milder 0.73% annual rate of decline, with a median age of 39.9 years old at start of mechanical ventilation. However, this analysis demonstrated that the average annual decline is dependent on age—faster decline in younger patients. Unfortunately, the effect of Nusinersen on the respiratory function in adult patients has been poorly studied so far. An encouraging 5% increase in the average Force Volume Capacity (FVC) of the lungs has been recently reported in Walter et al. [18]. Further studies need to be performed to comprehensively evaluate the impact of Nusinersen treatment on SMA adult population and to define the appropriate outcome measures for lung functional evaluation.

### 2.5. Additional Tests

Together with the tests discussed above, a series of other functional measures have been widely used to assess the electrophysiological activity of motor neurons, the axon number and reinnervation potential, and the structural changes in spinal cord and muscles. For example, electromyography (EMG), compound muscle action potential (CMAP), and motor neuron number index (MUNIX) have been classically used in patients affected with neuromuscular disorders to assess the functional status of the motor unit [41,42,43,44]. The EMG and CMAP methods have been used for SMA diagnosis [43,45,46] and the CMAP has also been demonstrated to be a good prognostic marker [47]. The scarcity of available data on the use of EMG and CMAP for the adult patient population has discouraged the attribution of these tests as ideal outcomes for adult SMA patients at the moment.

Computed from CMAP and electromyography interference pattern, the MUNIX is a quantitative test that aims to estimate the number of functional motor units in a specific muscle [44]. Overall, MUNIX and MUSIX are very interesting tests for the assessment of the number and size of motor units, which have been demonstrated to be directly correlated with motor function in adult type 2 and 3 patients [48]. However, no data is currently available on the reliability of this test to capture adult patients’ response to treatment (clinical trial NCT04139343 is currently recruiting patients to monitor MUNE in adults with SMA). Alterations in the cervical spinal cord of adult SMA patients have also been described due to a fine-tuned magnetic resonance imaging (MRI) [49] that can provide structural markers of the disease. Further refinement of the MRI techniques will certainly improve the sensitivity of the method [50] and apply it to a wider number of patients. It remains to be assessed whether it would be a reliable method to follow the response to treatments. Overall, the detailed evaluation of different functional outcomes gives insights regarding the disease progression. These efforts will contribute to a novel description of SMA patients, based on the classical definition of types and a more precise clinical assessment. However, observations made at the macro scale need to be complemented by molecular characterization in order to explain patient’s variability and device powerful predictive models of disease progression.

## 3. Genetic and Epigenetic Etiology of Clinical Heterogeneity

The discordance between genotype and phenotype among patients with same genetic mutation and same *SMN2* copy number can be explained by several factors, including (i) the presence of genetic modifiers and (ii) the epigenetic profile. These parameters need to be considered for both the natural history and to monitor the treatment response of SMA patients.

### 3.1. *SMN2* and Other Genetic Modifiers

Several genes have been identified in the last few years as SMA disease modifiers. These include Survival of Motor Neuron 2 (*SMN2*) [1,51], Plastin 3 (*PLS3*) [8], Coronin 1C (*CORO1C*) [9] and Neurocalcin Delta (*NCALD*) [10], Small EDRK-Rich Factor 1 (*SERF1*) [52], NLR Family Apoptosis Inhibitory Protein (*NAIP*) [53], General Transcription Factor IIH subunit 2 (i) [54], and Tolloid Like 2 (*TLL2*) [55]. Here, we will focus on the most documented genes.

The *SMN2* gene is the main modifier gene in SMA. Only 10% of the mRNA transcribed from the *SMN2* gene produces the functional FL-SMN protein. The number of copies of the *SMN2* genes is directly correlated to the level of functional FL-SMN and affects the severity of the disease (see Figure 1B) [1,7,47,51,56,57,58,59,60,61,62,63,64,65,66,67,68,69,70,71,72,73,74,75].

However, there are exceptions to this paradigm. Indeed, several type 2 and 3 SMA patients have been described to carry only two copies of *SMN2* instead of the expected three or four copies. This phenotype is often due to a rare single nucleotide variant (*SMN2* c.859G>C) in exon 7 that impacts *SMN2* gene splicing [11,76,77,78], inducing a higher rate of production of the FL-SMN protein (Figure 1B). Moreover, research from several groups described a variability of clinical phenotypes in siblings carrying the same *SMN1* mutation and identical *SMN2* copy number [5,58,79,80,81]. These events suggested the presence of modifier factors other than the *SMN2* gene are able to alter the severity of the disease and to be considered to define its full molecular signature.

### 3.2. Plastin3

F-actin bundling protein plastin 3 (*PLS3*) has been identified as the first gene, besides *SMN2*, to be able to attenuate the disease severity when upregulated (see Figure 1C). *PLS3* encodes a Ca^2+^-dependent F-actin-binding protein involved in neurotransmitter release and vesicle recycling at the presynaptic site [82]. The seminal work of Oprea et al. [8] demonstrated elevated levels of PLS3 in asymptomatic females of SMA type 2- or 3-affected sibling pairs where the males were always symptomatic. Each analyzed sibling pair carried identical homozygous *SMN1* deletion and the same number of *SMN2* copies. An additional study, described that the *PLS3* higher expression is maintained after generation and differentiation of induced pluripotent stem cells from fibroblasts of asymptomatic but not symptomatic siblings [83]. Unfortunately, no association of PLS3 expression was found in discordant female sibling pairs in an Iranian population [84]. Highly divergent results have also been shown about the role of PLS3 in mouse models of SMA. While two independent studies described an amelioration of the SMA phenotype upon overexpression of PLS3 in mice [82,85], another report showed that a randomly integrated PLS3 allele expressed in the severely affected Δ7-SMA mouse model failed to show rescue of the mice survival or motor function [86]. These discrepancies can be attributed to the fact that PLS3 is directly regulated by the SMN protein [87]; therefore, its overexpression is effective only when used as an adjuvant treatment to the classic SMN induction therapy [9]. It is plausible that the expression of PLS3 is affected by the SMA currently administered therapies and it could be used as a putative molecular biomarker.

### 3.3. PLS3-Interacting Protein CORO1C and CHP1

As a PLS3 binding partner, CORO1C regulates synaptic vesicles recycling in a calcium-dependent manner [9]. When overexpressed, *CORO1C* induced a significant amount of F-actin, therefore ameliorating endocytosis at the neuromuscular junctions [9,86] (see Figure 1C). Recently identified using a yeast-two-hybrid screen, the calcineurin-like EF-hand protein 1 (CHP1) is a novel PLS3 interacting protein [88] upregulated in SMA mice. Treatment of SMA mice with a low dose of AS oligonucleotide-SMN combined with *CHP1* downregulation improved the survival extension of the animals compared to the single SMN therapy [88]. The authors showed that the negative modulation of *CHP1* induced an activation of calcineurin with consequent restoration of endocytic protein phosphorylation. In the case where *CORO1C* and *CHP1* are differentially expressed in SMA patients, their modulation could also be beneficial to improve therapeutic effects.

### 3.4. NCALD

NCALD is calcium sensor protein suggested to negatively impact SMA via repression of endocytosis [10]. Reduced expression of NCALD was shown to be protective in a four-generation discordant family with five asymptomatic and two SMA1-affected individuals [10]. Moreover, reducing NCALD levels either in vitro or in vivo significantly ameliorated SMA pathology across SMA species [10]. Based on these encouraging results, Torres-Benito et al. [89] developed an ASO-based therapy to target *Ncald* in mouse spinal cord and used it in combination with low-dose SMN splice switching ASOs to design an efficient combinatorial therapy in SMA mice [89]. Therefore, repression of NCALD is considered protective for SMA.

### 3.5. NAIP

Another gene located in the same chromosomic region (5q13) of *SMN1* and *-2* is NLR Family Apoptosis Inhibitory Protein (*NAIP*). The functional role of NAIP in the pathogenesis of SMA has not been fully elucidated. However, some reports have demonstrated a correlation between deletion of the *NAIP* gene and severity of SMA [53,90] (see Figure 1). This gene was also found to be frequently deleted in 45% of the Egyptian patients [91]. Further investigation needs to be performed to validate the accuracy of this gene as a biomarker for SMA.

### 3.6. Epigenetic Modifiers (Methylation)

Epigenetic marks have been demonstrated to modulate the expression of the modifier genes in SMA. In particular, methylation in the promoter of the *SMN2* gene was reported to reduce its transcription and consequently the expression of the SMN protein, independently of the *SMN2* copy number (see Figure 1B). Studies on the methylation level of different CpGs on the *SMN2* genes revealed that hypomethylation was associated with reduced disease severity [92]. In fact, SMA type 3 exhibited a lower degree of methylation in the *SMN2* gene compared to type 1 or 2 patients [93]. After profiling the methylation in SMA patients and healthy controls, Zheleznyakova et al. identified differential degrees of methylation at CpG sites in the following genes; CHM Like Rab Escort Protein (*CHML*), Rho GTPase Activating Protein 22 (*ARHGAP22*), Cytokinesis And Spindle Organization B (*CYTSB*), Cyclin Dependent Kinase 2 Associated Protein 1 (*CDK2AP1*), and Solute Carrier Family 23 Member 2 (*SLC23A2*) [94]. Moreover, *SLC23A2* was significantly hypomethylated in type 3 and 4 patients compared to type 1 [95]. Thus, DNA methylation may regulate the SMA disease phenotype by modulating gene transcription and could be the molecular mechanism beyond the genotype–phenotype discrepancy often observed in SMA. Whether these changes are induced by environmental factors variations [96] and whether they are dependent or not on the absence of SMN, remain open questions.

### 3.7. Histone Deacetylases (HDAC) Inhibitors

Many efforts have also been made to target the HDACs and impact the transcription of the *SMN2* gene and its transduction into a functional SMN protein. Inhibition of HDACs impairs the removal of acetyl groups from the histone proteins and promotes gene transcription by mediating a more permissive, open, chromatin. In 2001, Chang et al. first identified an effect of a HDAC inhibitor, sodium butyrate, on SMA. Treatment with this epigenetic drug was able to increase the FL-SMN both in vitro and in vivo [97]. This discovery led to multiple investigations in the attempt to find more potent and stable HDAC inhibitors for SMA treatment. Valproic acid (VPA) [98,99], suberoylanilide hydroxamic acid (SAHA) [100,101], M344 [102], thricostatin A [103], and romidepsin [92] have also been suggested as promising treatments for SMA. The LBH589 molecule showed a higher potential than all other inhibitors, with the ability to induce a strong upregulation of SMN at very low doses [104]. Additionally, a combined therapy using the LBH589 inhibitor together with the Nusinersen drug showed a synergistic effect than the single treatments on SMA cellular models [105]. Despite the number of HDAC inhibitors tested and the encouraging results, only phenylbutyrate and VPA have entered clinical trials for human use. Valproic acid was tested in five Phase I/II clinical trials (NCT00374075, NCT00227266, NCT00481013, NCT00661453, and NCT01033331) [106,107,108,109,110,111] and showed outcomes variability. All the studies mentioned above were included in the meta-analysis performed by Elshafay et al. [112] suggesting the VPA administration was safe, although some adverse effects were recorded, and it induced a major improvement of motor function. The results of the SMA VALIANT trial (NCT00481013) on adult patients did not show any positive outcome on the motor function at either 6- or 12-month timepoints underlying that this treatment might have limited effects on the adult population [110]. A similar moderate success appeared from the pilot studies based on administration of phenylbutyrate to SMA patients [113,114,115]. More work needs to be done to establish whether these therapies could be used in combination with other approaches to efficiently ameliorate the SMA condition, even in the adults who seem less responsive to the currently approved therapies. The advantage of using HDAC inhibitors for SMA therapy, as other epigenetic drugs, is their ability to act on the chromatin without permanently affecting the DNA sequence. Moreover, some HDAC inhibitors have been demonstrated to reduce the methylation on the SMN2 promoter [92,101], therefore establishing a link between the two epigenetic modifications could be positively exploited in novel therapies.

## 4. Molecular Biomarkers

Biomarkers are needed in the field to provide the necessary insights to guide the decision-making in personalized medicine. There are several aspects to be considered in the identification of reliable biomarkers, like reproducibility and accuracy of measurements. We will discuss below the factors that have been identified as putative biomarkers for SMA.

### 4.1. SMN Protein

The SMN protein is considered the biomarker of choice for SMA, as all the approved therapies are aimed to restore its expression. SMN is ubiquitously expressed [116] and regulates several key processes in neuronal cells including ribonucleoprotein assembly, RNA metabolism [117], actin cytoskeleton dynamics [118], mRNA transport [119], ubiquitin homeostasis [120], bioenergetics pathways [121], synaptic vesicle release [122], and local protein translation [123,124,125]. Therefore, SMN-reduced expression in SMA has a devastating impact on many aspects of the neuronal homeostasis and survival. In fact, MNs expressing lower amounts of the SMN protein are more vulnerable to cell death [12] and this contributes to disease variability. The goal of the approved therapies for SMA is to re-express SMN in the affected tissues, mainly in MNs, but the scenario has become more complex since multiple studies published in the last few years have reported the importance of re-expressing SMN in peripheral tissues in addition to the CNS [126,127]. However, little is known on the specific tissue requirement for SMN; only recently a study from Ramos et al. described a variable level of SMN expression in different tissues and a general decline with aging [128]. Moreover, no robust correlations between *SMN2* copy number, and therefore SMA types, and SMN protein levels could have been demonstrated probably due to the tissue-specific SMN expression profile [70,128,129]. On the contrary, an analysis of SMN expression in spinal cord resident cells of treated patients revealed a higher amount of SMN protein in motor neurons after Nusinersen therapy [128]. These data suggest that SMN levels could serve as a potential biomarker to follow disease progression, although the reported discrepancies between SMN protein and mRNA expression [128,129] warrant more investigation. Additionally, as mentioned above, the tissue-specific requirement for SMN and its correlation with the patient’s response to treatment is largely unexplored and future research should address these problems to obtain a thorough follow-up, in particular for the adult SMA population.

### 4.2. Neurofilaments

Neurofilaments (NFs) are intermediate filaments of the neuronal axons, and they are abnormally released into the extracellular fluids, namely, CSF and peripheral blood, upon axonal damage in traumatic injuries or neurodegenerative disorders (reviewed in [130,131]). Constituted of three chains, light (NFl), medium (NF-M) and heavy (NF-H), based on their molecular weights, NFs are post-transcriptionally regulated [132]. In particular, the phosphorylation, abundant on the NF-H, ensures protection from degradation [133]. Although NFs seem to be reliable biomarkers for SMA infants, as their level is reduced upon Nusinersen treatment and this is generally correlated with an improved motor function [134], few recent publications have described contradicting results for the use of NFs as biomarkers for adult SMA patients’ response to therapy [29,135]. Eleven SMA type 3 patients (all adults—38.5 years mean age) analyzed for their pNF-H content in blood and CSF showed no change after administration of the Nusinersen loading doses [136]. In agreement with this effect, no significant difference was observed in the amount of NfL and pNF-H in another group of SMA patients (types 2 and 3) after the fourth injection of Nusinersen [135]. On the other hand, outcomes of a recently published analysis on a cohort of SMA type 3 patients (including kids, adolescents, and adults) showed a significant reduced amount of NFl and pNF-H after three injections with Nusinersen [29]. These discrepancies could be due to the different duration of follow-up, six months in the latter study against two months post-treatment for the previous ones. Technical limitations might also significantly contribute to the divergent outcomes (e.g., different sensitivity of the ELISA kits used). It is to be noted that at the analyzed time points, despite a decrease in the NFs levels, no locomotor function amelioration was observed in adult patients, suggesting that this aspect should be examined to a greater extent.

### 4.3. Protein Tau

Like neurofilaments, Tau is a neuron-specific structural protein, generally considered as a cortical neuronal marker. It was observed to be high in the CSF of patients with stroke [137], Alzheimer’s disease [138] and Amyothrophic lateral sclerosis (ALS) [139]. In SMA children, the baseline level of Tau in the CSF was significantly higher than in controls and it decreased with Nusinersen treatment [140]. On the contrary, the eleven SMA type 3 patients also analyzed for the NF content, described above by Totzeck et al. [136], did not exhibit alteration in the amount of tau protein. The analyses of tau protein and the discordant outcomes highlight once again the importance of defining bigger cohorts of patients, in particular adults, to study putative molecular biomarkers for SMA.

### 4.4. Serum Creatinine

Most biomarker studies published to date for SMA are focusing on neuronal-related molecules as previously discussed. If neurofilament and tau levels can mirror neuronal death, it is also known that the skeletal muscle metabolism is altered in neuromuscular disorders, and this could be another aspect to investigate in the search for molecular biomarkers. Indeed, due to either progressive denervation and also to the autonomous reduced level of SMN [141,142], metabolic activity of muscles is impaired as the disease progresses. Few biomarkers of muscle activity are known such as creatinine, a waste product from the phosphorylation of adenosine diphosphate by creatine that mostly occurs in skeletal muscles. Levels of creatinine are relatively lower in patients with SMA than controls [143], as reported in a longitudinal study of a 238 SMA patient cohort. Indeed, in these SMA patients, creatinine levels correlated closely with maximum CMAP and MUNE values, even after correction for confounding factors such age and lean mass (muscle atrophy). These data suggest that serum creatinine could constitute an interesting biomarker of the muscle deterioration and, more generally, of the disease progression. However, the study failed to provide insights regarding the intra-type correlation of creatinine levels and MUNE or CMAP outcomes. A fundamental problem, especially for type 3 patients, is the wide range of phenotype and a very variable disease progression. Additionally, the range of variability from visit to visit, which is what would be needed to guide therapeutic strategies, seems to be very wide. Therefore, additional studies should be performed in clinical trials to explore if the therapies would affect the creatinine levels, and if so, whether these differences would be distinguishable from natural variations in patients.

## 5. Strategies for the Discoveries of Novel Biomarkers Towards the Development of Personalized Medicine Approaches for SMA

The quest for biomarkers in SMA is still ongoing. Monitoring neurofilaments levels seems to be a very promising biomarker for the prognosis of treated infant patients [134], but does not appear to constitute a robust biomarker for adults patients. Indeed, studies including only adults patients showed comparable values between SMA patients and controls (see results published in [29]), suggesting that the variability observed in SMA samples could be unrelated to the SMA pathogenesis. Consequently, no prognosis biomarker for the outcome of therapies in adult patients is known to date.

Here, we review the strategies that could help tackle this problem, highlighting the approaches used in recent papers and based on the advantages of the unbiased -omics toolbox (Table 5). We then discuss perspectives on the use of these novel techniques for the future development of personalized medicine for SMA.

### 5.1. Multi-Omics Approaches for the Identification of SMA Biomarkers and Potential Therapeutic Targets

The first comprehensive -omics investigation of unprecedented scale and scope has been BforSMA (Biomarkers for SMA) [144]. The novelty of this study consisted on the use of several unbiased methods (metabolomics, proteomics, and transcriptomics) for the search of biomarkers in a large well-defined SMA patient cohort and age-matched healthy controls. Despite the identification of several putative biomarkers, further validation is needed to confirm these findings. The field is now widely relying on the use of unbiased approaches to highlight potential molecular candidates. Transcriptomic and proteomic approaches are our best hope to grasp the complex mechanism that shapes the landscape of SMA clinical heterogeneity. A genome-wide RNA sequencing analysis on MNs differentiated from patient-derived induced pluripotent stem cells (iPSC) revealed that endoplasmic reticulum stress is upregulated in SMA, representing a novel potential target pathway [145]. A similar transcriptomic analysis on iPSC has been used to identify enriched motifs in differentially expressed genes between SMA and control cells, and pinpoints synaptogamin binding cytoplasmic RNA interacting protein (SYNCRIP) as a key modulator of SMN [146].

Furthermore, an interesting strategy published by Nizzardo et al. [147] took advantage of the known pathophysiological processes in motor neuron disorders (MND) to perform a comprehensive transcriptomic analysis. In particular, the authors started from the observation that specific types of MNs are spared during the progression of the MND, like SMA and ALS. They then compared the transcriptomic profile of affected MNs of the brainstem and spinal cord with the rather unaffected ocular MNs in ALS [147]. This approach led to the identification of several differentially expressed genes (DEGs), including synaptotagmin, which was demonstrated to be a neuroprotective protein in ALS. These results, also confirmed in SMA patient-derived MNs, opened new avenues for the investigation of potential novel biomarkers for SMA. In an independent study, a proteomic approach has been used to screen for candidate proteins regulated by the Nusinersen treatment in a cohort of ten type 2 and 3 patients [30]. Different proteomic clusters were identified containing proteins that are differentially expressed before and after six Nusinersen injections. This analysis allowed the identification of neuronal and non-neuronal proteins that could not only be valuable biomarkers but they could also be putative targets of SMA combined therapies.

### 5.2. Perspectives for Personalized Medicine in Neuromuscular Disorders

Successful examples of personalized medicine approaches relying on multi-omics tools come from the cancer field. Indeed, a great effort in the last decade was focused in designing targeted therapies based on cancer mutation profiles [148]. The combination of molecular screening and analysis of predictive biomarkers represents a successful emerging strategy for cancer treatment [148]. Similarly, for high-burden bacterial infections, next-generation sequencing (NGS) of the patient’s fluids will surely revolutionize the diagnostic process. The pathogen can be easily identified thanks to available bioinformatic data analyzed via artificial intelligence (AI) workflows and, consequently, the appropriate therapy can be quickly determined [149,150]. For SMA and other neuromuscular disorders, these approaches could ideally lead to personalized therapies but major hurdles need to be overcome. First, for these diseases the affected tissues are not readily available for biopsy and, second, the disease spectrum is highly complex, as thoroughly discussed above [151]. Thus, further endeavors are necessary to define personalized medicine approaches for the broad SMA population.

## 6. Conclusions and Discussion

The complexity of the SMA phenotypes, and the discrepancies between patients’ genotype (as per current definition) and phenotype, have fostered the search for functional and molecular biomarkers that could help to better classify patient types. The discovery of several disease modifier genes, starting with *SMN2* [1], and epigenetic factors [92] has revolutionized the traditional classification and account for a wider complexity. This aspect will further change in light of the recently approved therapies that are modifying the course of the disease and increasing patients’ survival. For the functional measurements, it is now clear that the available tests to date are inappropriate to grasp the small changes in locomotor abilities or electrophysiological parameters of adult SMA patients. In fact, these changes seem to fall into a different scale compared to the ones form younger patients that have been widely investigated. Therefore, refining the existing scales and finding novel functional measures is a priority [152]. On the other hand, many efforts have been devoted to the identification of reliable molecular biomarkers but all the proposed candidates have revealed some limitations. Therefore, novel screening methods, approaches for an accurate prognosis, and biomarkers for treatment follow-up are being identified. Recent technological advances have allowed the development of many tools that will considerably improve the extent of the analysis. For example, novel sensitive techniques such as NGS can currently identify a broader range of genetic and epigenetic differences that can be explored as possible biomarkers. Therefore, further investigation needs to be performed to unbiasedly identify putative biomarkers for accurate diagnosis, prognosis, and treatment monitoring in SMA. These efforts, combined with refined AI approaches, will represent a milestone for a successful personalized medicine development in SMA. 

## Figures and Tables

**Figure 1 jpm-10-00075-f001:**
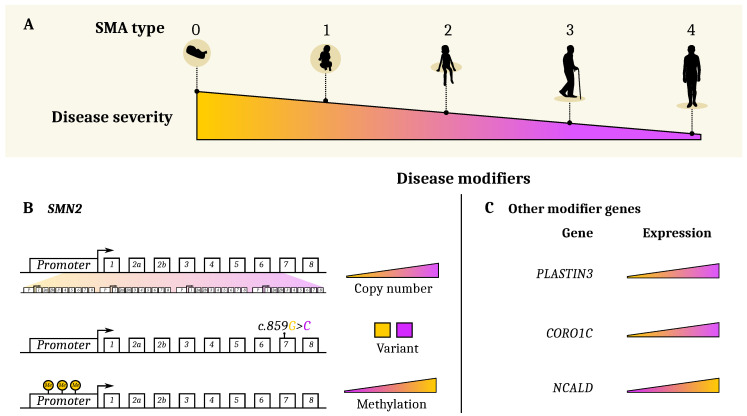
**Known molecular disease modifiers.** (**A**) Current classification of SMA as a discrete projection of a continuous spectrum of disease severity. This spectrum results from the combination of (**B**) disease modifiers such as *SMN2* copy number, *SMN2* variant, and methylation state; (**C**) level of expression of *CORO1C* and *PLASTIN3* was associated with milder phenotype, while *NCALD* overexpression was associated with more severe phenotype.

**Table 1 jpm-10-00075-t001:** **Current classification of spinal muscular atrophy (SMA) patients in 5 types.** The subtypes, age of onset (mo.: months; y.: years), level of motor functions, life expectancy, and frequency of *SMN2* copy number (*SMN2*cn) are reported [7]. Background colors are used to differentiate SMA types.

SMA Type	Subtype	Age of Onset	Level of Motor Functions	Life Expectancy	*SMN2*cn (%)
0/1a		pre-natal	Need respiratory assistance	<1 month	
1		0–6 mo.	Cannot sit independently	<2 years	2 (73.4%)
	1b		Absence of head control and ability to roll over		
	1c		Sometimes gain head control or the ability to roll from supine to prone position		
2		<18 mo.	Cannot stand independently	>2 years	3 (81.8%)
	2a		Independent sitting lost		
	2b		Independent sitting conserved		
3		>18 mo.	Able to stand and walk independently	Adulthood	3–4 (50.6%; 45.5%)
	3a	18 mo.–3 y.			
	3b	>3 y.			
4		>20 y.	Weaknesses in lower limbs	Adulthood	

**Table 2 jpm-10-00075-t002:** **Summary of representative Hammersmith Functional Motor Scale Expanded (HFMSE) scores by SMA subtype.** Graphical representation of the HFMSE scores in SMA patients from five independent studies. Columns are named as follows. **References**: bibliographic citations of the natural history or Nusinersen-related studies. **Cohort**: cohort composition in SMA patients with specified type or subtype when the information is available. **T**: time when the analysis was performed; 0: baseline; +10 mo: 10 months after first Nusinersen dose. **HMSFE**: scores represented as a boxplot; values are indicated as median ± standard deviation (s.d.). Δ HMFSE ($T-T_{0}$) indicates the increment score between the baseline before treatment ($T_{0}$) and the final after treatment timepoint (*T*). **N**: number of patients included. **Age**: age of the subject in years as reported in the relative study (median (s.d.)). Colors of boxplot and dots are related to the cohort characteristics (SMA type): mustard: SMA type 2; green: type 3; light green: subtype 3b. Background color associates consecutive table entries referring to the same study.

References	Cohort	T	HFMSE	N	Age
*Natural history studies*			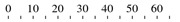		
Kaufmann et al. [27]	type 2	/	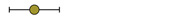	41	9.1 (7.4)
	type 3	/	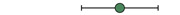	38	13.7 (10.8)
Montes et al. [31]	type 2	/	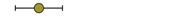	67	10.9 (8.3)
	type 3	/	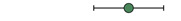	59	13.4 (10.7)
Faravelli et al. [29]	type 3	/	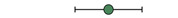	12	29 (15–35)
Walter et al. [18]	type 3	/	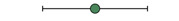	19	29 (15–35)
Kessler et al. [30]	type 3b	0	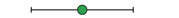	7	39 (13)
			Δ HFMSE ($T-T_{0}$)		
*Nusinersen-related studies*			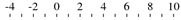		
Kessler et al. [30]	type 3b	+10 mo.	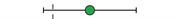	7	39 (13)
Hagenacker et al. [19]	type 2	+10 mo.	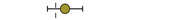	30
	type 3	+10 mo.	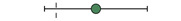	60	37 (12)

**Table 3 jpm-10-00075-t003:** **Summary of representative results for the Revised Upper Limb Module (RULM).** Graphical representation of the RULM scores in SMA patients from three independent studies. Columns are named as follows. **References**: bibliographic citations of the natural history or Nusinersen-related studies. **Cohort**: cohort composition in SMA patients with specified type or subtype. **T**: time when the analysis was performed; 0: baseline; +10 mo: 10 months after first Nusinersen dose. **RULM**: scores represented as a boxplot; values are indicated as median ± sd. Δ RULM ($T-T_{0}$) indicates the increment score between the baseline before treatment ($T_{0}$) and the final after treatment timepoint (T). **N**: number of patients included. **Age**: age of the subject in years as reported in the relative study (age range); NA: not available. Colors of boxplot and dots are related to the cohort characteristics (SMA type): mustard: SMA type 2; green: type 3. Background color associates consecutive table entries referring to the same study.

References	Cohort	T	RULM	N	Age
*Natural history studies*			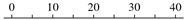		
Stolte et al. [36]	type 2	/	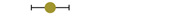	9	24 to 48
	type 3	0	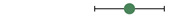	19	18 to 61
Walter et al. [18]	type 3	0	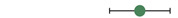	19	18 to 59
			Δ RULM ($T-T_{0}$)		
*Nusinersen-related study*			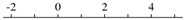		
Hagenacker et al. [19]	type 2	+10 mo.	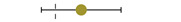	30	*NA*
	type 3	+10 mo.	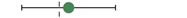	58	*NA*

**Table 4 jpm-10-00075-t004:** **Summary of representative results for the 6-Minute-Walk-Test (6MWT) by ambulatory patients.** Graphical representation of the 6MWT in healthy and SMA patients from four independent studies. Columns are named as follows. **References**: bibliographic citations of the natural history or Nusinersen-related studies. **Cohort**: cohort composition in healthy and/or SMA patients with specified type or subtype when the information is available. **T**: time when the analysis was performed. **Distance**: distance reported in meters and represented as a boxplot; values are indicated as median ± standard deviation (s.d.). Δ Distance ($T-T_{0}$) indicates increment distance covered between the baseline before treatment ($T_{0}$) and the final after treatment timepoint (T). **N**: number of patients included. **Age**: age of the subject in years as reported in the relative study (age range or mean (s.d.)); NA: not available. Colors of boxplot and dots are related to the cohort characteristics (SMA type or controls): gray: healthy subject; green: SMA type 3; dark green: SMA subtype 3a; light green: SMA subtype 3b. Background color associates consecutive table entries referring to the same study.

References	Cohort	T	Distance (m)	N	Age
*Natural history studies*			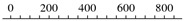		
Mckay et al. [40]	Healthy	/	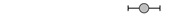	400	20 to 59
Montes et al. [28]	type 3a	0	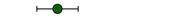	57	10.3 (9.8)
	type 3b	0	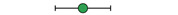	28	25.6 (12.5)
			Δ Distance ($T-T_{0}$)		
			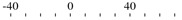		
Montes et al. [28]	type 3a	+1 y	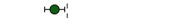	57	10.3 (9.8)
	type 3b	+1 y	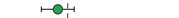	28	25.6 (12.5)
*Nusinersen-related studies*			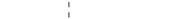		
Hagenacker et al. [19]	type 3	+6 mo.	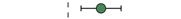	47	*NA*
		+10 mo.	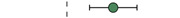	37	*NA*
		+14 mo.	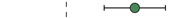	25	*NA*

**Table 5 jpm-10-00075-t005:** **Overview of multi-omics approaches used to date to characterize SMA and its progression. References:** bibliographic citations of the -omics studies. **-Omics:** Specific techniques employed in each study. **Samples’ source:** biological material analyzed. **Highlights:** main insights on SMA pathogenesis gained in each study. Background colors are used to differentiate the cited studies.

References	-Omics	Samples’ Source	Highlights
Finkel et al. [144]	Metabolomic, transcriptomic, proteomic	Plasma and urine from 108 SMA patients type 1, 2 and 3 (between 2 and 12 years of age)	97 proteins and 59 metabolites in the plasma together with 44 metabolites in the urine correlated with functional score
Rizzo et al. [146]	Transcriptomic	iPSCs-derived motorneurons from SMA patients and healthy controls	NRXN2 protein downregulation was identified as potentially neuroprotective.
Nizzardo et al. [147]	Transcriptomic	Spinal and ocular motoneurons isolated from human central nervous system sections from MND patients	Synaptogamin13 was identified as a putative neuroprotective protein in MND.
Kessler et al. [30]	Proteomic	CSF samples from 10 Nusinersen-treated adults SMA type 2 and 3	No correlation between protein profiling and functional score evolution, over 10 months treatment

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
