# Peer review of "The Identification of Novel Biomarkers Is Required to Improve Adult SMA Patient Stratification, Diagnosis and Treatment"

_jpm, 2020, doi:10.3390/jpm10030075_

Round 1

Reviewer 1 Report

The manuscript by Smeriglio et al, titled as ‘Understanding the Functional and Molecular Features of the Heterogeneous Adult SMA Patient Population for Precision Medicine’ has valuable content that may be considered for publication.

The following specific comments may be of help to improve the manuscript.

Title:  The title is incomplete. It requires rewriting.

Abstract: Overall, it is good.

The following sentence requires rewriting:

Attempts to capture phenotypic differences10 and to understand the biological signature, i.e. the use of novel -omics approaches, are needed to11 revise the disease classification and device personalized treatments.

Key words: Good

Introduction:

1.The highlighted part of the sentence may need correction:

‘Spinal muscular atrophy (SMA) is a neurodegenerative disease affecting motoneurons (MN) in16 the brainstem and the spinal cord caused by the homozygous disruption of the survival of motor17 neuron 1 (SMN1) gene’

Is it homozygous mutation causing disruption……?

  1. The following sentence states that there are 5 different types, while in the abstract the authors mention about 4 subtypes:

The disease presents in a wide spectrum of clinical severity and patients18 are classified in 5 types, depending on the age of onset and motor milestones achieved.

The authors may want to correct it.

  1. The following sentence needs a reference: ‘Attempts to explain this phenotypic22 variability have been made and several disease modifiers are known’.
  2. The following sentence is incomplete: ‘The most widely accepted is23 the copy number of the SMN2 gene, a centromeric paralog of SMN1’
  3. Rewrite the following sentence: ‘Therefore, the higher the SMN2 copy number, the stronger is the SMN protein expression and 28 milder is the phenotype ‘
  4. Rewrite the following sentence: ‘Moreover, therapeutic monitoring seems to be even more challenging for 55 adult patients with SMA types 1 subtype c – 4 due to the lack of natural history data, the different56 rate of muscle function deterioration and the higher heterogeneity of this patient population heterogeneity of this patient population’
  5. Rewrite the following sentence by splitting into two sentences: In this context,the identification of novel biomarkers and precise methods for functional outcomes 60 evaluation are unmet needs to i) redefine patients’ classification, ii) characterize the full molecular 61 profile of therapy responders or non-responders and iii) determine inclusion criteria in clinical trials62 and treatment protocols’

Segment on ‘Evaluation of functional outcomes in adult patient’:

Good

Table 1 : Good

Table 2: Good

Table 3: Good

Table 4: Good

Segment on, ‘Genetic and epigenetic etiology of clinical heterogeneity’: Good

1.The following sentence require rewriting: The discordance between genotype-phenotype and among phenotypes in patients with identical223 background and genetic defects can be explained by several factors including……..

Figure 1 : Good

Segment on, ‘Molecular biomarkers’ : Good

Segment on, ‘Strategies for the discoveries of novel biomarkers: Good

Segment on, ‘The -omics approach for the SMA biomarkers future search’: This segment is disappointing. The authors are encouraged to add a small table summarizing the key findings from these ‘omics technology. This effort will add strength to this timely article.

A summary paragraph on personalized medical approaches will add strength. The authors should outline the information about those approaches that worked well and those that did not work. They may add details about which subgroup showed successful outcome with what personalized approach.

Author Response

The manuscript by Smeriglio et al, titled as ‘Understanding the Functional and Molecular Features of the Heterogeneous Adult SMA Patient Population for Precision Medicine’ has valuable content that may be considered for publication.

We are thankful to the reviewer for considering the manuscript suitable for publication. All the comments were appreciated and we modified the text accordingly. We have also added a table for the -omics paragraph and added a section on the personalized medicine approaches as suggested by the reviewer.

  • Title

   The title is incomplete. It requires rewriting.

We have now changed the title to: The identification of novel biomarkers is required to improve adult SMA patient stratification, diagnosis and treatment 

  • Abstract:

Rewriting of the following sentence: Attempts to capture phenotypic differences10 and to understand the biological signature, i.e. the use of novel -omics approaches, are needed to11 revise the disease classification and device personalized treatments.

We thank the reviewer for having noticed the mistake in this sentence. We have now changed it to: “We finally highlight the need for novel techniques, i.e. -omics approaches, to capture phenotypic differences and to understand the biological signature in order to revise the disease classification and device personalized treatments.

  • Introduction:

  1. Correction of the following sentence: ‘Spinal muscular atrophy (SMA) is a neurodegenerative disease affecting motoneurons (MN) in the brainstem and the spinal cord caused by the homozygous disruption of the survival of motor neuron 1 (SMN1) gene’

We agreed with the reviewer. The sentence has been corrected to: “Spinal muscular atrophy (SMA) is a neurodegenerative disease affecting motoneurons (MN) in the brainstem and spinal cord caused by the homozygous mutation of the survival of motor neuron 1 (SMN1) gene’

  1. Correction of the following sentence: The disease presents in a wide spectrum of clinical severity and patients are classified in 5 types, depending on the age of onset and motor milestones achieved.

We thank the reviewer for highlighting this discrepancy. As described in Table 1, there are 5 SMA types. Therefore, we have changed this information in the abstract to be consistent with the text and table. The text in the abstract now appears as follows: “Spinal muscular atrophy (SMA), is currently classified into five different subtypes, from the most severe (type 0) to the mildest (type 4) depending on age at onset, best motor function achieved and copy number of the SMN2 gene.

  1. The following sentence needs a reference: Attempts to explain this phenotypic variability have been made and several disease modifiers are known’

As suggested by the reviewer, we have added the following reference for this sentence (Maretina M.A., Current Genomics, 2018, DOI: 10.2174/1389202919666180101154916).

  1. The following sentence is incomplete: ‘The most widely accepted is the copy number of the SMN2 gene, a centromeric paralog of SMN1’

We have changed the sentence to: “The most widely accepted modifier is the copy number of the SMN2 gene, a centromeric paralogue of SMN1”.

  1. Rewrite the following sentence: ‘Therefore, the higher the SMN2 copy number, the stronger is the SMN protein expression and milder is the phenotype‘

The sentence has been changed to: “SMN2 copy number is directly correlated with the SMN expression level; therefore, a higher copy number is associated with a milder phenotype. Therefore, SMN2 copy number…”

  1. Rewrite the following sentence: ‘Moreover, therapeutic monitoring seems to be even more challenging for adult patients with SMA types 1 subtype c – 4 due to the lack of natural history data, the different rate of muscle function deterioration and the higher heterogeneity of this patient population’

Upon the reviewer’s suggestion we have rephrased this sentence as follows: “Moreover, due to the lack of natural history data and the higher heterogeneity among the adult SMA patient population, therapeutic monitoring of these patients is particularly challenging”.

  1. Rewrite the following sentence by splitting into two sentences: In this context, the identification of novel biomarkers and precise methods for functional outcomes evaluation are unmet needs to i) redefine patients’ classification, ii) characterize the full molecular profile of therapy responders or non-responders and iii) determine inclusion criteria in clinical trials and treatment protocols’.

We thank the reviewer for this suggestion. This is the modified sentence: “In this context, the identification of novel biomarkers and precise methods for functional outcomes evaluation are unmet needs. Reaching these goals will provide means to i) redefine patients’ classification, ii) characterize the full molecular profile of therapy responders or non-responders and iii) determine inclusion criteria in clinical trials and treatment protocols. Therefore, several types of biomarkers…”.

Segment: Genetic and epigenetic etiology of clinical heterogeneity’

  1. The following sentence require rewriting: The discordance between genotype-phenotype and among phenotypes in patients with identical background and genetic defects can be explained by several factors including……..

We have changed the sentence to: “The discordance between genotype and phenotype among patients with same genetic mutation and same SMN2 copy number can be explained by several factors including…”

Segment: The -omics approach for the SMA biomarkers future search

  1. This segment is disappointing. The authors are encouraged to add a small table summarizing the key findings from these ‘omics technology. This effort will add strength to this timely article.

We thank the reviewer for this comment. We have modified the paragraph on the omics tools by discussing few more recent publications that we believe are relevant and add strength to the manuscript. Moreover, as suggested, we have prepared a table (Table 5) to summarize the key findings from the use of the omics techniques. We believe that this paragraph is now more solid and comprehensive.

  1. A summary paragraph on personalized medical approaches will add strength. The authors should outline the information about those approaches that worked well and those that did not work. They may add details about which subgroup showed successful outcome with what personalized approach.

We agreed with the reviewer’s suggestion on adding a conclusive paragraph about personalized medicine approaches. We have added the text as paragraph 5.2. before the final conclusions and discussion. In this paragraph, after discussing few successful examples of personalized medicine approaches from cancer and infectious diseases, we described the current hurdles that need to be overcome in the neuromuscular field to apply the latest technology towards personalize medicine treatments for SMA patients. 

Reviewer 2 Report

This article provides a comprehensive and meaningful overview of SMA molecular biomarkers, functional evaluations, and future research for developing precision medication. It is well written and will provide a solid foundation for many future research studies. I commend the authors on a well-done narrative review.

I have no major comments, just a few small grammatical changes. I would suggest one more thorough revision for language prior to the final submission.

Page 1, Line 34. Please pluralize siblings

Page 13. Line 457. Please remove the hyphen from refining.

Page 13. Line 458-460. This sentence is confusing as written. Please rewrite.

Author Response

This article provides a comprehensive and meaningful overview of SMA molecular biomarkers, functional evaluations, and future research for developing precision medication. It is well written and will provide a solid foundation for many future research studies. I commend the authors on a well-done narrative review.

We thank the reviewer for appreciating the content and style of our review. We have corrected the grammar mistakes as suggested.

  • Page 1, Line 34. Please pluralize siblings

We have now changed sibling into siblings

  • Page 13. Line 457. Please remove the hyphen from refining.

We have changed re-fining into refining

  • Page 13. Line 458-460. This sentence is confusing as written. Please rewrite.

We thank the reviewer for highlighting the mistake. We have changed the sentence to : “On the other hand, many efforts have been devoted to the identification of reliable molecular biomarkers but all the proposed candidates have revealed some limitations.

Reviewer 3 Report

Authors revised the current methods for functional evaluation of adult spinal muscular atrophy (SMA) patients treated with Nusinersen, and the known molecular changes underpinning disease heterogeneity. Besides, they discussed the possible strategies for the discovery of novel biomarkers. The addressed topic is interesting, and the manuscript is well organized.

The following minor issue should be addressed:

  • On page 8, line 237, a typing error should be amended ("there are exception").

Author Response

Authors revised the current methods for functional evaluation of adult spinal muscular atrophy (SMA) patients treated with Nusinersen, and the known molecular changes underpinning disease heterogeneity. Besides, they discussed the possible strategies for the discovery of novel biomarkers. The addressed topic is interesting, and the manuscript is well organized.

We are grateful to the reviewer to consider our manuscript clear, interesting and well organized. We have incorporated the suggested change as outlined below.

On page 8, line 237, a typing error should be amended ("there are exception").

Thanks to the reviewer for pointing out the grammar mistake. We have changed “there are exception” into “there are exceptions”.